# Air Sampling for Fungus around Hospitalized Patients with Coronavirus Disease 2019

**DOI:** 10.3390/jof8070692

**Published:** 2022-06-30

**Authors:** Yi-Chun Chen, Yin-Shiou Lin, Shu-Fang Kuo, Chen-Hsiang Lee

**Affiliations:** 1Division of Infectious Diseases, Department of Internal Medicine, Kaohsiung Chang Gung Memorial Hospital, Kaohsiung 83301, Taiwan; sonice83@yahoo.com.tw (Y.-C.C.); shiou0428@cgmh.org.tw (Y.-S.L.); 2Department of Laboratory Medicine, Kaohsiung Chang Gung Memorial Hospital, Kaohsiung 83301, Taiwan; ivykuo@cgmh.org.tw; 3Department of Medical Biotechnology and Laboratory Sciences, College of Medicine, Chang Gung University, Taoyuan 33302, Taiwan; 4School of Medicine, College of Medicine, Chang Gung University, Taoyuan 33302, Taiwan

**Keywords:** fungus, air sampling, COVID-19, invasive aspergillosis

## Abstract

The risk of developing coronavirus disease 2019 (COVID-19)-associated pulmonary aspergillosis (CAPA) depends on factors related to the host, virus, and treatment. However, many hospitals have modified their existing rooms and adjusted airflow to protect healthcare workers from aerosolization, which may increase the risk of *Aspergillus* exposure. This study aimed to quantitatively investigate airborne fungal levels in negative and slightly negative pressure rooms for COVID-19 patients. The air in neutral pressure rooms in ordinary wards and a liver intensive care unit with high-efficiency particulate air filter was also assessed for comparison. We found the highest airborne fungal burden in recently renovated slightly negative air pressure rooms, and a higher airborne fungal concentration in both areas used to treat COVID-19 patients. The result provided evidence of the potential environmental risk of CAPA by quantitative microbiologic air sampling, which was scarcely addressed in the literature. Enhancing environmental infection control measures to minimize exposure to fungal spores should be considered. However, the clinical implications of a periodic basis to determine indoor airborne fungal levels and further air sterilization in these areas remain to be defined.

## 1. Introduction

Patients with severe coronavirus disease 2019 (COVID-19) are at high risk of fungal infections due to its association with epithelial lung damage, lymphopenia, and dysfunction of the cell immune response, as well as the use of broad-spectrum antibiotics, corticosteroids, and immunosuppressive agents [1,2]. Invasive aspergillosis is an important complication in patients with severe COVID-19 pneumonia, and it is associated with poor outcomes [3,4]. The reported incidence of COVID-19-associated pulmonary aspergillosis (CAPA) varies due to heterogeneity in patient populations, monitoring protocols, diagnostic tools, and definitions, and a review study of 37 articles reported a pooled incidence of 10.0% [5]. Diagnosis of CAPA can be challenging owing to absence of typical host factors, non-specific radiological findings [6], and difficulty in obtaining mycologic evidence. Many atypical signs of COVID-19 pneumonia can mimic invasive pulmonary aspergillosis and vice versa [7]. Given the large number of COVID-19 cases requiring critical care management globally, CAPA represents a significant burden of disease and considerable mortality (more than 50%) [3,8,9].

Inhalation of airborne *Aspergillus* conidia is critical for infection [10], and therefore increased exposure through local environmental conditions is an important issue. These conditions include construction of new hospitals or units, repurposing of noncritical care areas to provide care for COVID-19 patients, and potentially the use of negative pressure rooms, which have been associated with an increase in the concentration of *Aspergillus* conidia. Contaminated respiratory equipment (humidifiers, nebulizers, oxygen cannisters) has also been investigated as a potential source of fungi in healthcare settings [5]. However, few studies have quantitatively assessed airborne fungal levels in hospital rooms with COVID-19 patients. Ichai et al. reported that 6 out of 26 patients developed CAPA and 2 patients were colonized by *Aspergillus fumigatus* in a negative pressure intensive care unit (ICU). Analysis of the air in three of the four rooms was positive for *A. fumigatus*. Switching to a slightly positive room pressure combined with standard environmental cleaning protocols successfully eradicated *Aspergillus* from the air in these rooms, and no new cases of CAPA or *Aspergillus* sp. colonization were reported for up to at least three months. These findings suggest that implementing negative pressure in an ICU room could lead to air contamination by *Aspergillus*, and thus increase the risk of opportunistic infections [11]. To identify potential sources of contamination and determine the necessity of regular air quality control, we conducted this study to quantitatively investigate airborne fungal levels in negative pressure rooms and slightly negative pressure rooms that were recently constructed to care for COVID-19 patients. The air in neutral pressure rooms in ordinary wards and liver intensive care unit (LICU) rooms with high-efficiency particulate air (HEPA) filter was also assessed for comparison. Comparing microbiologic sampling results from a target area (an area of care for COVID patients) to those in the ordinary wards in the facility could provide information about concentration of potential airborne pathogens and environmental risk to development of CAPA, which was scarcely addressed before this study.

## 2. Materials and Methods

### 2.1. Air Sampling

Environmental air samples were collected at Kaohsiung Chang Gung Memorial Hospital (KCGMH) between January and March 2022. The samples were collected in 12 negative pressure rooms, 22 slightly negative air pressure rooms for COVID-19 patients, 46 neutral pressure rooms in two ordinary wards, and 14 LICU rooms with HEPA filter for solid organ transplantation recipients. There is an anteroom in the negative pressure rooms and the minimum differential pressure between isolation room and ambient pressure is 8 Pa. There is no anteroom in the slightly negative pressure rooms, but an exhaust fan was equipped in the window. In the neutral pressure rooms, the door can be opened freely. There are 6 positive pressure (≥+2.45 Pa) and 8 neutral pressure rooms in the LICU. The room temperature and humidity in negative pressure rooms, slightly negative pressure rooms, neutral pressure rooms in ordinary wards, and rooms in a LICU during air sampling were 23 °C, 19 °C, 18–22 °C, and 21 °C and 90%, 73%, 58–73%, and 60%, respectively. The Hyper Light Disinfection Robot, model: Hyper Light P3 (Mediland Enterprise Corporation, Taoyuan, Taiwan, R.O.C) is a mobile, automatic device, which is made for environmental disinfection by UV-C irradiation (254 nm) [12]. The device was used routinely in the LICU rooms after patients were discharge or transfer from rooms. The airborne fungal levels from one LICU room were investigated before and immediately after environmental disinfection with UV-C irradiation for 30 min.

The samples were collected using a MAS-100NT Microbial Air Sampler (Merck Millipore, Darmstadt, Germany). One air sample was obtained from each room. During collection, the air sampler was placed in the middle of the room at approximately 1.5 m above floor level. A total of 500 L [13] of air was collected directly onto malachite green agar 2.5 ppm plates. The plates were incubated at 25 °C and read on Day 7.

### 2.2. Data Analyses

If any suspected *A. fumigatus* colonies, which have the characteristics, including velvety, downy, or powdery surface growth, showing various shades of green, most commonly a blue-green to a grey-green with a narrow white border and white to tan to pale yellowish reverse [14,15], were noted, 1 to 3 colonies were chosen for confirmation using conventional identification methods based on morphological characteristics and matrix-assisted laser desorption/ionization time-of-flight mass spectrometry with a biotyper, v3.1 (Microflex LT, Bruker Daltonik GmbH, Bremen, Germany). Cell Counter, a plugin for ImageJ software v1.4.3.67 (an open-source and widely used image processing tool developed by the National Institute of Health, Bethesda, MD, USA) was used to count the total fungal and *A. fumigatus*-like colonies.

### 2.3. Statistical Analyses

*p*-values were determined using the Student’s *t* test or Mann–Whitney U test to compare the number of colony forming units (CFU) of total fungal and *A. fumigatus*-like colonies in the four hospital areas. All statistical tests were two-tailed, and significance was set at α = 0.05. Analyses were performed using SPSS v17.0 for Windows (SPSS Inc., Chicago, IL, USA).

## 3. Results

The median numbers of total fungal colonies in the air of negative pressure rooms, slightly negative pressure rooms, neutral pressure rooms, and LICU rooms were 67 (interquartile range (IQR) 53.3–100), 161 (IQR 124.8–197.8), 26.5 (IQR 18.8–43.0), and 4.5 (IQR 0–8.8) CFU per 500 L air, respectively. The median numbers of *A. fumigatus*-like colonies in the air of negative pressure rooms, slightly negative pressure rooms, neutral pressure rooms in the ordinary wards and LICU rooms with HEPA filter were 46 (35.8–58.5) (IQR), 107 (71.5–149.8), 12.5 (7.0–20.3), and 0 (0–2.5) CFU per 500 L air, respectively (Figure 1). Among these areas, the slightly negative pressure rooms had the significantly highest airborne fungal level. Air fungal burden in the negative pressure rooms was significantly higher than that in the neutral pressure rooms (*p* < 0.001), and air fungal burden in the neutral pressure rooms was significantly higher than that in the LICU rooms (*p* < 0.001). Figure 2 shows a representative agar plate inoculated with fungi from the air in the four areas. Figure 2D,E shows agar plates cultured with air from one LICU room before and after environmental disinfection with ultraviolet-C (UV-C) irradiation (254 nm) using Hyper Light Disinfection Robot for 30 min. The number of fungal colonies decreased from 15 to 2 per 500 L air before and after UV-C disinfection, respectively.

## 4. Discussion

In the present study, we found that slightly negative pressure rooms had the highest airborne fungal concentration, followed by negative pressure rooms. A possible explanation for these findings is that some construction was conducted in the slightly negative pressure rooms before receiving COVID-19 patients. In addition, compared with rooms with negative pressure, there was no built-in ultraviolet light disinfection in these slightly negative pressure rooms. There were higher airborne fungal concentrations in both areas arranged for COVID-19 patients than in the ordinary wards. This raise concerns that patients, and in particular those who are immunocompromised, treated in these isolated rooms may have an increased risk of exposure to *Aspergillus*. Environmental surveillance of these areas, especially in rooms that were modified for COVID-19 patients, should be enhanced.

Several outbreaks of environmental airborne fungal infections within a hospital setting have been reported, in which more than 90% of the patients were immunocompromised [16,17]. At present, there is no uniform definition of nosocomial aspergillosis. One reason for the difficulty in defining hospital-acquired aspergillosis is that the incubation period of invasive aspergillosis is unknown [18]. Generally, invasive disease that occurs after 1 week of hospitalization is considered nosocomial [18]. The median time of onset of CAPA has been reported to be 8.0 days after ICU admission [9], corresponding to later coinfection than in influenza patients, in whom it generally occurs only 2 days after ICU admission [19]. According to this timeline, there may be a link between hospital environmental exposure and the risk of developing CAPA.

This study has several limitations. To date, no patient has been diagnosed with CAPA at KCGMH. Therefore, we were unable to demonstrate a relationship between clinical and hospital environmental *Aspergillus* isolates. We were also unable to confirm the identification of every colony. Therefore, we could only report *A. fumigatus*-like colonies, and misreading of the colonies may have occurred. That might be one of reasons why higher percentage of *A. fumigatus*-like colonies were revealed in the present study than previous report [20]. Alberti C et al. proposed that other fungal species can serve as good indirect markers of a higher risk of invasive nosocomial aspergillosis [20]. Although most of these molds are non-pathogenic, their detection showed a lack of effective filtration or cleaning and/or the existence of conditions that favor the settling of molds, including *Aspergillus* [20]. Furthermore, culture isolation focuses on the presence of particular fungi, and molecular methods such as next generation sequencing, are reliable tools for identifying and tracking the fungal diversity of indoor air [21]. Finally, some variables that could have influenced the results of air samples including seasons, weather, and work progress [22] were not considered in this study. The strength of the study is that we used volumetric and quantitative methods to determine the fungal burden in indoor air. Although the minimal airborne concentration of *Aspergillus* spores necessary to cause infection in patients with significant immunodeficiency remains unknown, increasing the frequency of air treatment by portable UV-C disinfection to decrease air fungal load is feasible. Further investigations are necessary to demonstrate the efficacy of regular air quality control in reducing the incidence of CAPA.

## 5. Conclusions

We found the highest airborne fungal burden in slightly negative air pressure rooms that were recently renovated, and a higher airborne fungal concentration in both areas used to treat COVID-19 patients. The result provided evidence of the potential environmental risk of CAPA by quantitative microbiologic air sampling, which was scarcely addressed in previous studies. Enhancing environmental infection control measures to minimize exposure to fungal spores should be considered. However, the clinical implications of a periodic basis to determine indoor airborne fungal levels and further air sterilization in the areas for care of patients with severe COVID-19 remain to be defined.

## Figures and Tables

**Figure 1 jof-08-00692-f001:**
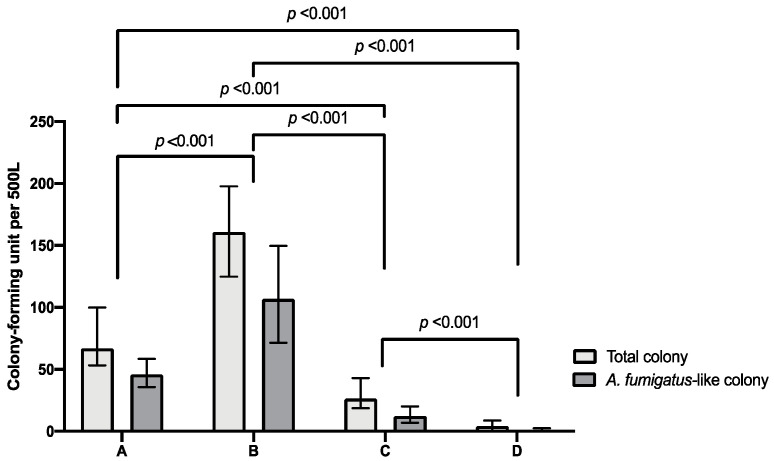
The median and interquartile range (IQR) of total fungal and *A. fumigatus*-like colonies in the air of different hospital environments. Negative pressure rooms (A), slightly negative pressure rooms (B), neutral pressure rooms in ordinary wards (C), and rooms in a liver intensive care unit with high-efficiency particulate air filter (D). The highest air fungal burden, either total fungal colonies or *A. fumigatus*-like colonies, was found in the slightly negative pressure rooms (B). In addition, there were higher airborne fungal concentrations in both negative and slightly negative pressure rooms (A,B) arranged for COVID-19 patients than the rooms in the ordinary ward (C) and liver intensive care unit with high-efficiency particulate air filter (D).

**Figure 2 jof-08-00692-f002:**
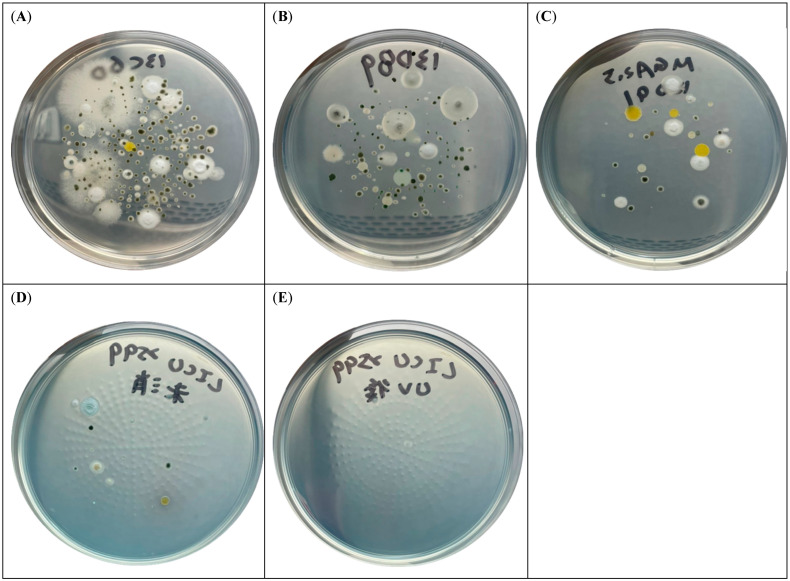
Representative agar plates inoculated by airborne fungi from the different hospital environments. Slightly negative air pressure rooms which had recently been renovated (**A**). Negative air pressure rooms (**B**). Rooms in the ordinary ward (**C**). Before and after environmental disinfection with ultraviolet-C (UV-C) irradiation (254 nm) in a liver intensive care unit room (**D**,**E**).

## Data Availability

Data are available on request.

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
