# Peer review of "Air Sampling for Fungus around Hospitalized Patients with Coronavirus Disease 2019"

_jof, 2022, doi:10.3390/jof8070692_

Round 1

Reviewer 1 Report

In this paper, the objective of the authors was to analyze the impact of negative and slightly negative pressure rooms on the fungal aerobiocontamination compared to neutral pressure and rooms equipped with HEPA filters. 

This paper doesn’t analyze the risk factor for invasive aspergillosis in COVID-19 patients but provides evidence of the environmental risk, a topic in which we lack of data.

Please find some comments in order to optimize the manuscript.

Materials and methods:

- What data base was used to identify filamentous fungi with Maldi-ToF?

- Was antifungal susceptibility testing performed to detect azole resistant Aspergillus?

- Please specify the weather during this study: rainy? dry? temperatures?

- What is the definition of Aspergillus-like fungi because the rate of these fungi compared to the total flora is quite high. Were all filamentous fungi counted here? or only “green” filamentous colonies?

Results:

- Because the authors are skilled in monitoring environmental biocontamination, do they have baseline values at least in the neutral pressure wards and in rooms equipped with HEPA filters?

- Figure 2 legend. Please specify A to E (probably lines 108-111 in the main text)

Discussion:

- The authors should discuss that 1 cubic meter is the usual sampling volume in rooms equipped with HEPA filtration and explain that here it is only 500 L to be comparable with other measures.

- What are the impacts of this study for the authors? Did they switch from negative to neutral pressure? Did they implement cleaning measures?

- Open the discussion on new molecular techniques that can be used for environmental monitoring.

References:

Some references could be added regarding:

- sampling methods in hospitals:

Méheust D, Le Cann P, Reboux G, Millon L, Gangneux JP. Indoor fungal contamination: health risks and measurement methods in hospitals, homes and workplaces. Crit Rev Microbiol. 2014 Aug;40(3):248-60. doi: 10.3109/1040841X.2013.777687. Epub 2013 Apr 16. PMID: 23586944.

- Total fungal flora is representative of the presence of Aspergillus:

Alberti C, Bouakline A, Ribaud P, Lacroix C, Rousselot P, Leblanc T, Derouin F; Aspergillus Study Group. Relationship between environmental fungal contamination and the incidence of invasive aspergillosis in haematology patients. J Hosp Infect. 2001 Jul;48(3):198-206. doi: 10.1053/jhin.2001.0998. PMID: 11439007.

- New techniques of environmental monitoring 

Gangneux JP, Sassi M, Lemire P, Le Cann P. Metagenomic Characterization of Indoor Dust Bacterial and Fungal Microbiota in Homes of Asthma and Non-asthma Patients Using Next Generation Sequencing. Front Microbiol. 2020 Jul 30;11:1671. doi: 10.3389/fmicb.2020.01671. PMID: 32849345; PMCID: PMC7409152.

Author Response

Air sampling for fungus around hospitalized patients with coronavirus disease 2019 (Manuscript number: jof-1786168)

Response to the Reviewers

Reviewer #1:

In this paper, the objective of the authors was to analyze the impact of negative and slightly negative pressure rooms on the fungal aerobio-contamination compared to neutral pressure and rooms equipped with HEPA filters. 

This paper doesn’t analyze the risk factor for invasive aspergillosis in COVID-19 patients but provides evidence of the environmental risk, a topic in which we lack of data.

Please find some comments in order to optimize the manuscript.

Materials and methods:

What data base was used to identify filamentous fungi with Maldi-ToF?

Response: Thank you for raising the question. Matrix‐assisted laser desorption/ ionization time‐of‐flight mass spectrometry with a biotyper, version 3.1 (MALDI‐TOF MS; Bruker Daltonics) was used to identify filamentous fungi. (page 2-3, line 98-99)

Was antifungal susceptibility testing performed to detect azole resistant Aspergillus?

Response: Thank you for raising the question. The objective of the study was to analyze the impact of negative and slightly negative pressure rooms on the fungal aerobio-contamination compared to neutral pressure and rooms equipped with HEPA filters. As a result, antifungal susceptibility testing was not performed in the present study. However, we have investigated azole resistance of Aspergillus spp. from hospital environments (Chen YC, Kuo SF, Wang HC, Wu CJ, Lin YS, Li WS, Lee CH. Azole resistance in Aspergillus species in Southern Taiwan: an epidemiological surveillance study. Mycoses. 2019;62:1174-81). We found low prevalence of azole resistance of Aspergillus spp. in clinical and environmental isolates.

Please specify the weather during this study: rainy? dry? temperatures?

Response: Thanks for your suggestion. We recorded the temperature and humidity during air sampling. We add the description “Room temperature and humidity in negative pressure rooms, slightly negative pressure rooms, neutral pressure rooms in ordinary wards and rooms in a LICU during air sampling was 23°C, 19°C, 18-22°C and 21°C and 90%, 73%, 58-73% and 60%, respectively. (page 2, line 78-80)

What is the definition of Aspergillus-like fungi because the rate of these fungi compared to the total flora is quite high. Were all filamentous fungi counted here? or only “green” filamentous colonies?

Response: Thank you for pointing this out. We revise the sentence “If any suspected A. fumigatus colonies, which have the characteristics, including velvety, downy or powdery surface growth, showing various shades of green, most commonly a blue-green to a grey-green with a narrow white border or white to tan to pale yellowish reverse [14,15],”. (page 2, line 93-96)

According to the characteristics, we counted the total fungal and A. fumigatus-like colonies separately. In the part of limitations, we revise the description, “We were also unable to confirm the identification of every colony. Therefore, we could only report A. fumigatus-like colonies, and misreading of the colonies may have occurred. That might be one of reasons why higher percentage of A. fumigatus-like colonies were revealed in the present study than previous report [20]”. (page 5, line 164-168)

Results:

Because the authors are skilled in monitoring environmental biocontamination, do they have baseline values at least in the neutral pressure wards and in rooms equipped with HEPA filters?

Response: Thanks for your comment. There were no outbreaks of environmental airborne fungal infection in our hospital before the current study. Periodic air quality monitoring has not been implemented. Therefore, baseline data of these hospital environments was unavailable.

Figure 2 legend. Please specify A to E (probably lines 108-111 in the main text)

Response: Thanks for your suggestion. We add the type of room in the caption of Figure 2.

Discussion:

The authors should discuss that 1 cubic meter is the usual sampling volume in rooms equipped with HEPA filtration and explain that here it is only 500 L to be comparable with other measures.

Response: Thanks for your opinion. Troiano G et al. conducted a fungal surveillance during demolition activities in a healthcare facility after extraordinary preventive measures and reported that the total volume of sampled air ranged from 240 to 390 L in the wards (Troiano, G.; Sacco, C.; Donato, R.; et al. Demolition activities in a healthcare facility: results from a fungal surveillance after extraordinary preventive measures. Public Health 2019, 175, 145-147). Sampling volumes are usually adapted to the level of fungal concentration. Therefore, they are higher in hospitals (usually 500 L or 1 m3) with low presumed levels of biocontamination than in workplaces where lower volumes will avoid saturating the sample by impaction (usually <300 L) (Meheust, D.; Le Cann, P.; Reboux, G.; et al. Indoor fungal contamination: health risks and measurement methods in hospitals, homes and workplaces. Crit Rev Microbiol 2014, 40, 248-260). The standard sampling volumes of the air sampler used in this study (MAS‐100NT Microbial Air Sampler [Merck Millipore]) were 50, 100, 250, 500 and 1000L according to manufacturer ‘s recommendations. In our preliminary data, higher volumes of air collection such as 1000L lead the colonies in the agar plates too busy to be counted. We finally determined that the total collected volume of sampled air was 500 L.

What are the impacts of this study for the authors? Did they switch from negative to neutral pressure? Did they implement cleaning measures?

Response: Thanks for your comment. For admission of COVID-19 patients, we did not switch negative pressure to neutral pressure. Therefore, air sterilization by a mobile, automatic device, which is made for environmental disinfection by UV-C irradiation (254 nm) has been considered. The effectiveness of the UV-C devices in eradicating airborne fungi in these hospital environments will be addressed in the future. We presented the results of UV-C disinfection in a LICU room (Figure 2D-E).

Open the discussion on new molecular techniques that can be used for environmental monitoring.

Response: Thanks for your suggestion. We add the description, “Furthermore, culture isolation which focuses on the presence of particular fungi, molecular methods such as next generation sequencing are reliable tools for identifying and tracking the fungal diversity of indoor air [21]”. (page 5, line 172-174)

References:

Some references could be added regarding:

- sampling methods in hospitals:

Méheust D, Le Cann P, Reboux G, Millon L, Gangneux JP. Indoor fungal contamination: health risks and measurement methods in hospitals, homes and workplaces. Crit Rev Microbiol. 2014 Aug;40(3):248-60. doi: 10.3109/1040841X.2013.777687. Epub 2013 Apr 16. PMID: 23586944.

- Total fungal flora is representative of the presence of Aspergillus:

Alberti C, Bouakline A, Ribaud P, Lacroix C, Rousselot P, Leblanc T, Derouin F; Aspergillus Study Group. Relationship between environmental fungal contamination and the incidence of invasive aspergillosis in haematology patients. J Hosp Infect. 2001 Jul;48(3):198-206. doi: 10.1053/jhin.2001.0998. PMID: 11439007.

- New techniques of environmental monitoring 

Gangneux JP, Sassi M, Lemire P, Le Cann P. Metagenomic Characterization of Indoor Dust Bacterial and Fungal Microbiota in Homes of Asthma and Non-asthma Patients Using Next Generation Sequencing. Front Microbiol. 2020 Jul 30;11:1671. doi: 10.3389/fmicb.2020.01671. PMID: 32849345; PMCID: PMC7409152.

Response: We appreciate the important references you provided. We cite them as new Ref 13, 20 and 21. (page 2, line 90; page 5, line 166-172; page 5, line 172-174)

Reviewer 2 Report

The presented work is an interesting study about the potentially increased risk of pulmonary aspergillosis in negative pressure rooms, and this topic could easily be considered as current and of interest considering covid-19 patients are commonly located in negative pressure rooms.

Still I think some changes are necessary:

-       Materials and methods

o   (Lines 61-65) No indication is provided about the differential pressures recorded for the rooms in this study. What was the difference in numbers/ranges of values between “negative pressure”, “slighty negative”, “neutral” and the LICU rooms?

o   (Line 71) I think more information is needed about the morphological or other type of criteria used to define a colony as “suspected” for A.fumigatus

o   (Lines 73-75) I would think that the versions of Cell Counter plugin and ImageJ software used in this study should be stated. It should also be specified if the plugin was used to count all the colonies and then the A.fumigatus-like colonies were noted separately, or if the software could also distinguish between the two possibilities.

o   (Line 76) A parametric and a non-parametric test are reported. It is necessary to report which was it that it was used, for which set of data, and based on which criteria. Was a normality test involved, and what were the results?

o   No information about the UV-C irradiation in the involved rooms is reported in this section: this subject appears only in the Results and Discussion section. It is necessary to report what were the differences in the UV-C disinfection in all the rooms involved, and the timing of it with the air sampling.

-       Results

o   Figure 1

§  I think it would be better to state the p-values in the image or the caption

§  Interquartile range is not visible in the full black bars. It would be better if a lighter shade of grey was used, or some pattern to make the inferior limit of IQR always visible for all bars

o   Figure 2

§  caption is below a portion of text in the manuscript.

§  Also, I think it would be better if the correspondence between the letter and the type of room would also be in the image or the caption

-       Discussion

o   (Line 115) What type of construction was conducted, and how much time before the sampling? A starting point for some variables that could have influenced the results can be found in:

§  Troiano G, Sacco C, Donato R, Pini G, Niccolini F, Nante N. Demolition activities in a healthcare facility: results from a fungal surveillance after extraordinary preventive measures. Public Health. october 2019;175:145–7.

o   (Lines 116-117) Information about the UV-C irradiation in the involved rooms, stated in the Discussion section, is missing in the Methods section

Author Response

Air sampling for fungus around hospitalized patients with coronavirus disease 2019 (Manuscript number: jof-1786168)

Response to the Reviewers

Reviewer #2:

The presented work is an interesting study about the potentially increased risk of pulmonary aspergillosis in negative pressure rooms, and this topic could easily be considered as current and of interest considering covid-19 patients are commonly located in negative pressure rooms.

Still I think some changes are necessary:

Materials and methods

(Lines 61-65) No indication is provided about the differential pressures recorded for the rooms in this study. What was the difference in numbers/ranges of values between “negative pressure”, “slightly negative”, “neutral” and the LICU rooms?

Response: Thank you for pointing this out. We add descriptions about the pressures recorded in these four areas as “There is an anteroom in the negative pressure rooms and the minimum differential pressure between isolation room and ambient pressure is 8 Pa. There is no anteroom in the slightly negative pressure rooms but an exhaust fan was equipped in the window. In the neutral pressure rooms, the door can be opened freely. There is 6 positive pressure (≥ +2.45 Pa) and 8 neutral pressure rooms in the LICU”. (page 2, line 73-77)

(Line 71) I think more information is needed about the morphological or other type of criteria used to define a colony as “suspected” for A. fumigatus

Response: Thanks for your comment. We revise the sentence as “If any suspected A. fumigatus colonies, which have the characteristics, including velvety, downy or powdery surface growth, showing various shades of green, most commonly a blue-green to a grey-green with a narrow white border and white to tan to pale yellowish reverse [14,15],”. (page 2, line 93-96)

(Lines 73-75) I would think that the versions of Cell Counter plugin and ImageJ software used in this study should be stated. It should also be specified if the plugin was used to count all the colonies and then the A. fumigatus-like colonies were noted separately, or if the software could also distinguish between the two possibilities.

Response: Thanks for your opinion. We add the version of ImageJ software used in this study (page 3, line 100). The plugin was only used to count the total colonies and A. fumigatus-like colonies. It was not used to differentiate between A. fumigatus-like colonies and non-A. fumigatus-like colonies.

(Line 76) A parametric and a non-parametric test are reported. It is necessary to report which was it that it was used, for which set of data, and based on which criteria. Was a normality test involved, and what were the results?

Response: Thanks for your opinion. A normality test has been performed. According to Kolmogorov-Smirnov test, data set of negative pressure rooms and slightly negative pressure rooms belonged to normal distribution. Student’s t test was used to compare the number of colony forming units in these two areas while other comparisons were performed by Mann–Whitney U test.

No information about the UV-C irradiation in the involved rooms is reported in this section: this subject appears only in the Results and Discussion section. It is necessary to report what were the differences in the UV-C disinfection in all the rooms involved, and the timing of it with the air sampling.

Response: Thanks for your suggestion. We add the information about UV-C disinfection as (page 2, line 80-86), “The Hyper Light Disinfection Robot, model: Hyper Light P3 (Mediland Enterprise Corporation, Taoyuan, Taiwan, R.O.C) is a mobile, automatic device, which is made for environmental disinfection by UV-C irradiation (254 nm) [12]. The device was used routinely in the LICU rooms after patients were discharged or transferred from the rooms. The airborne fungal levels from one LICU room was investigated before and immediately after environmental disinfection with UV-C irradiation for 30 minutes.”

Results

Figure 1

I think it would be better to state the p-values in the image or the caption

Response: Thank you for your suggestion. We add the p-values in the Figure 1.

Interquartile range is not visible in the full black bars. It would be better if a lighter shade of grey was used, or some pattern to make the inferior limit of IQR always visible for all bars

Response: Thank you for your suggestion. We revise the Figure 1.

Figure 2 caption is below a portion of text in the manuscript.

Response: Thank you for your suggestion. In the revised edition, Figure 2 is put below a portion of text in the manuscript.

Also, I think it would be better if the correspondence between the letter and the type of room would also be in the image or the caption

Response:

Thank you for your suggestion. We add the type of room in the caption of Figure 2.

Discussion

(Line 115) What type of construction was conducted, and how much time before the sampling? A starting point for some variables that could have influenced the results can be found in: Troiano G, Sacco C, Donato R, Pini G, Niccolini F, Nante N. Demolition activities in a healthcare facility: results from a fungal surveillance after extraordinary preventive measures. Public Health. october 2019;175:145–7.

Response: Thank you for your opinion. Air sampling was performed 2 months after renovation. The conducted construction for slightly negative pressure rooms included modification of airflow by equipment of exhaust fan and enclosure of the nursing station.

We add the description “Finally, some variables that could have influenced the results of air samples including seasons, weather, and work progress were not considered in this study” in the limitations and cite the reference. (page 5, line 174-176)

(Lines 116-117) Information about the UV-C irradiation in the involved rooms, stated in the Discussion section, is missing in the Methods section

Response: Thank you for your opinion. We add information about UV-C irradiation in the section of Materials and methods. (page 2, line 80-86)

Reviewer 3 Report

1) Abstract:  L15- 23. The risk of developing coronavirus disease 2019 (COVID-19)-associated pulmonary as-  pergillosis depends on factors related to host, virus and treatment. However, many hospitals have  modified their existing rooms and adjusted airflow to protect healthcare workers from aerosolization, which may increase the risk of Aspergillus exposure. This study aimed to quantitatively investigate airborne fungal levels in negative and slightly negative pressure rooms for COVID-19 patients. The air in neutral pressure rooms in ordinary wards and a liver intensive care unit with high efficiency particulate air filter was also assessed for comparison. We found the highest air fungal  burden in recently renovated slightly negative air pressure rooms. Monitoring air quality and controlling environmental risk factors to minimize exposure to fungal spores should be considered. However, the clinical implications remain to be defined. Please the abstract is quite rumbling. Could you please divide it in different sections such as background, aim, conclusions ..

2) 1. Introduction L26-31. Patients with severe coronavirus disease 2019 (COVID-19) are at high risk of fungal  infections due to its association with epithelial lung damage, lymphopenia, dysfunction  of the cell immune response, and the use of broad-spectrum antibiotics, corticosteroids and immunosuppressive agents [1,2]. Invasive aspergillosis is an important complication in patients with severe COVID-19 pneumonia, and it is associated with poor outcomes [3]. Please ameliorate this paragraph and add these references:

A- Epidemiology of Invasive Pulmonary Aspergillosis Among Intubated Patients With COVID-19: A Prospective Study. Clin Infect Dis. 2021;73(11):e3606-e3614. doi:10.1093/cid/ciaa1065

B- Potential links between COVID-19-associated pulmonary aspergillosis and bronchiectasis as detected by high resolution computed tomography. Front Biosci (Landmark Ed). 2021;26(12):1607-1612. doi:10.52586/5053

c- Different Methods to Improve the Monitoring of Noninvasive Respiratory Support of Patients with Severe Pneumonia/ARDS Due to COVID-19: An Update. J Clin Med. 2022;11(6):1704. Published 2022 Mar 19. doi:10.3390/jcm11061704

3) 1. Introduction L52-58. These findings suggest that implementing negative pressure in an ICU room  could lead to air contamination by Aspergillus, and thus increase the risk of opportunistic infections [8]. To identify potential sources of contamination and determine the necessity  of regular air quality control, we conducted this study to quantitatively investigate airborne fungal levels in negative pressure rooms and slightly negative pressure rooms which had recently been constructed to care for COVID-19 patients. The air in neutral  pressure rooms in ordinary wards and liver intensive care unit (LICU) rooms with high- 58 efficiency particulate air (HEPA) filter was also assessed for comparison. Please underline the novelty of the study and the clinical implications.

4) 2. Materials and methods L60-79.  Environmental air samples were collected at Kaohsiung Chang Gung Memorial Hospital (KCGMH) between January and March 2022. The samples were collected in 12 neg- ative pressure rooms, 22 slightly negative air pressure rooms for COVID-19 patients, 46 neutral pressure rooms in two ordinary wards, and 14 LICU rooms with HEPA filter for  solid organ transplantation recipients. The samples were collected using a MAS‐100NT  Microbial Air Sampler (Merck Millipore). One air sample was obtained from each room. During collection, the air sampler was placed in the middle of the room at approximately  1.5 m above floor level. A total of 500 L of air was collected directly onto malachite green  agar 2.5 ppm plates. The plates were incubated at 25°C and read on Day 7. If any suspected  A. fumigatus colonies were noted, 1 to 3 colonies were chosen for confirmation using conventional identification methods based on morphological characteristics and matrix-as-sisted laser desorption/ionization time–of–flight mass spectrometry (Microflex LT, Bruker  Daltonik GmbH, Bremen, Germany). Cell Counter, a plugin for ImageJ software (an open-source and widely used image processing tool developed by the National Institute of  Health) was used to automatically analyze the total fungal and A. fumigatus-like colonies. P-values were determined using the Student’s t test or Mann–Whitney U test to compare  the number of colony forming units (CFU) of total fungal and A. fumigatus-like colonies in  the four hospital areas. All statistical tests were 2-tailed, and significance was set at α =  0.05. Analyses were performed using SPSS v17.0 for Windows (SPSS Inc., Chicago, IL, USA). The paragraph is quite rumbling and difficult to read.

Could you please divide it in different sections (e.g. data collection, statistical analysis, …)?

5) 5. Conclusions L145-149.  We found the highest airborne fungal burden in slightly negative air pressure rooms  which had recently been renovated, and a higher airborne fungal concentration in both areas used to treat COVID-19 patients than in ordinary wards. Enhancing environmental  infection control measures in these areas to minimize exposure to fungal spores should be considered. However, the clinical implications remain to be defined.  Please underline the novelty of the study and the clinical implications.

Author Response

Air sampling for fungus around hospitalized patients with coronavirus disease 2019 (Manuscript number: jof-1786168)

Response to the Reviewers

Reviewer #3:

1) Abstract:  L15- 23. The risk of developing coronavirus disease 2019 (COVID-19)-associated pulmonary aspergillosis depends on factors related to host, virus and treatment. However, many hospitals have modified their existing rooms and adjusted airflow to protect healthcare workers from aerosolization, which may increase the risk of Aspergillus exposure. This study aimed to quantitatively investigate airborne fungal levels in negative and slightly negative pressure rooms for COVID-19 patients. The air in neutral pressure rooms in ordinary wards and a liver intensive care unit with high efficiency particulate air filter was also assessed for comparison. We found the highest air fungal  burden in recently renovated slightly negative air pressure rooms. Monitoring air quality and controlling environmental risk factors to minimize exposure to fungal spores should be considered. However, the clinical implications remain to be defined. Please the abstract is quite rumbling. Could you please divide it in different sections such as background, aim, conclusions.

Response: Thank you for your opinion. We would like to divide the abstract into different sections as below:

Background: The risk of developing coronavirus disease 2019 (COVID-19)-associated pulmonary aspergillosis depends on factors related to host, virus and treatment. However, many hospitals have modified their existing rooms and adjusted airflow to protect healthcare workers from aerosolization, which may increase the risk of Aspergillus exposure.

Methods: This study aimed to quantitatively investigate airborne fungal levels in negative and slightly negative pressure rooms for COVID-19 patients. The air in neutral pressure rooms in ordinary wards and a liver intensive care unit with high efficiency particulate air filter was also assessed for comparison.

Results: We found the highest air fungal burden in recently renovated slightly negative air pressure rooms.

Conclusions: Monitoring air quality and controlling environmental risk factors to minimize exposure to fungal spores should be considered. However, the clinical implications remain to be defined.

However, according to author guidelines of Journal of fungi, The abstract should be a single paragraph and should follow the style of structured abstracts, but without headings: 1) Background: Place the question addressed in a broad context and highlight the purpose of the study; 2) Methods: Describe briefly the main methods or treatments applied. Include any relevant preregistration numbers, and species and strains of any animals used. 3) Results: Summarize the article's main findings; and 4) Conclusion: Indicate the main conclusions or interpretations.”

Therefore, we suggest to maintain the original style of abstract in the manuscript.

2) 1. Introduction L26-31. Patients with severe coronavirus disease 2019 (COVID-19) are at high risk of fungal infections due to its association with epithelial lung damage, lymphopenia, dysfunction of the cell immune response, and the use of broad-spectrum antibiotics, corticosteroids and immunosuppressive agents [1,2]. Invasive aspergillosis is an important complication in patients with severe COVID-19 pneumonia, and it is associated with poor outcomes [3]. Please ameliorate this paragraph and add these references:

A- Epidemiology of Invasive Pulmonary Aspergillosis Among Intubated Patients With COVID-19: A Prospective Study. Clin Infect Dis. 2021;73(11):e3606-e3614. doi:10.1093/cid/ciaa1065

B- Potential links between COVID-19-associated pulmonary aspergillosis and bronchiectasis as detected by high resolution computed tomography. Front Biosci (Landmark Ed). 2021;26(12):1607-1612. doi:10.52586/5053

c- Different Methods to Improve the Monitoring of Noninvasive Respiratory Support of Patients with Severe Pneumonia/ARDS Due to COVID-19: An Update. J Clin Med. 2022;11(6):1704. Published 2022 Mar 19. doi:10.3390/jcm11061704

Response: Thank you for your suggestion. We do our best to ameliorate this paragraph and add the references as new Ref 4, 6, 7. (page 1, line 32; page 1, line 35-36; page 1, line 36-38)

3) 1. Introduction L52-58. These findings suggest that implementing negative pressure in an ICU room could lead to air contamination by Aspergillus, and thus increase the risk of opportunistic infections [8]. To identify potential sources of contamination and determine the necessity of regular air quality control, we conducted this study to quantitatively investigate airborne fungal levels in negative pressure rooms and slightly negative pressure rooms which had recently been constructed to care for COVID-19 patients. The air in neutral pressure rooms in ordinary wards and liver intensive care unit (LICU) rooms with high- efficiency particulate air (HEPA) filter was also assessed for comparison. Please underline the novelty of the study and the clinical implications.

Response: Thank you for your suggestion. We rewrite the paragraph as "To identify potential sources of contamination and determine the necessity of regular air quality control, we conducted this study to quantitatively investigate airborne fungal levels in negative pressure rooms and slightly negative pressure rooms which had recently been constructed to care for COVID-19 patients. The air in neutral pressure rooms in ordinary wards and liver intensive care unit (LICU) rooms with high-efficiency par-ticulate air (HEPA) filter was also assessed for comparison. Comparing microbiologic sampling results from a target area (an area of care for COVID patients) to those in the ordinary wards in the facility could provide information about concentration of potential airborne pathogens and environmental risk to development of CAPA, which had been scarcely addressed." to underline the novelty of the study and the clinical implications. (page 2, line 57-66)

4) 2. Materials and methods L60-79.  Environmental air samples were collected at Kaohsiung Chang Gung Memorial Hospital (KCGMH) between January and March 2022. The samples were collected in 12 negative pressure rooms, 22 slightly negative air pressure rooms for COVID-19 patients, 46 neutral pressure rooms in two ordinary wards, and 14 LICU rooms with HEPA filter for solid organ transplantation recipients. The samples were collected using a MAS‐100NT Microbial Air Sampler (Merck Millipore). One air sample was obtained from each room. During collection, the air sampler was placed in the middle of the room at approximately 1.5 m above floor level. A total of 500 L of air was collected directly onto malachite green agar 2.5 ppm plates. The plates were incubated at 25°C and read on Day 7. If any suspected A. fumigatus colonies were noted, 1 to 3 colonies were chosen for confirmation using conventional identification methods based on morphological characteristics and matrix-assisted laser desorption/ionization time–of–flight mass spectrometry (Microflex LT, Bruker Daltonik GmbH, Bremen, Germany). Cell Counter, a plugin for ImageJ software (an open-source and widely used image processing tool developed by the National Institute of Health) was used to automatically analyze the total fungal and A. fumigatus-like colonies. P-values were determined using the Student’s t test or Mann–Whitney U test to compare the number of colony forming units (CFU) of total fungal and A. fumigatus-like colonies in the four hospital areas. All statistical tests were 2-tailed, and significance was set at α = 0.05. Analyses were performed using SPSS v17.0 for Windows (SPSS Inc., Chicago, IL, USA). The paragraph is quite rumbling and difficult to read.

Could you please divide it in different sections (e.g. data collection, statistical analysis, …)?

Response: Thank you for your suggestion. We divide the text in Materials and Methods into three sections as “Air Sampling, Data Analyses, and Statistical Analyses”.

5) 5. Conclusions L145-149.  We found the highest airborne fungal burden in slightly negative air pressure rooms which had recently been renovated, and a higher airborne fungal concentration in both areas used to treat COVID-19 patients than in ordinary wards. Enhancing environmental infection control measures in these areas to minimize exposure to fungal spores should be considered. However, the clinical implications remain to be defined.  Please underline the novelty of the study and the clinical implications. 

Response: Thank you for your suggestion. We rewrite the paragraph as "We found the highest airborne fungal burden in slightly negative air pressure rooms which had recently been renovated, and a higher airborne fungal concentration in both areas used to treat COVID-19 patients. The result provided evidence of the potential environmental risk of CAPA by quantitative microbiologic air sampling which had been scarcely addressed. Enhancing environmental infection control measures in these areas to minimize exposure to fungal spores should be considered. However, the clinical implications of a periodic basis to determine indoor airborne fungal levels and further air sterilization in the areas for care of patients with severe COVID-19 remain to be defined." to underline the novelty of the study and the clinical implications. (page 5, line 184-191)